**Data Availability Statement:** The raw data are provided in S1 Table.

**Funding:** This work was supported by Monash University and the World Mosquito Program,

# Extensive variation and strain-specificity in dengue virus susceptibility among African *Aedes aegypti* populations

Stéphanie Dabo[1], Annabelle Henrion-Lacritick[2], Alicia Lecuyer[1], Davy Jiolle[3,4], Christophe Paupy[3,4], Diego Ayala[3,4], Silvânia da Veiga Leal[5], Athanase Badolo[6], Anubis Vega-Rúa[7], Massamba Sylla[8], Jewelna Akorli[9], Sampson Otoo[9], Joel Lutomiah[10], Rosemary Sang[10], John-Paul Mutebi[11], Maria-Carla Saleh[2], Noah H. Rose[12,13¤], Carolyn S. McBride[12,13], Louis Lambrechts [1]*

**1** Institut Pasteur, Université Paris Cité, CNRS UMR2000, Insect-Virus Interactions Unit, Paris, France, **2** Institut Pasteur, Université Paris Cité, CNRS UMR3569, Viruses and RNA Interference Unit, Paris, France, **3** MIVEGEC, Montpellier University, IRD, CNRS, Montpellier, France, **4** Centre Interdisciplinaire de Recherches Médicales de Franceville, Franceville, Gabon, **5** Laboratório de Entomologia Médica, Instituto Nacional de Saúde Pública, Praia, Cabo Verde, **6** Laboratoire d'Entomologie Fondamentale et Appliquée, Université Joseph Ki-Zerbo, Ouagadougou, Burkina Faso, **7** Institut Pasteur of Guadeloupe, Laboratory of Vector Control Research, Transmission Reservoir and Pathogens Diversity Unit, Morne Jolivière, Guadeloupe, France, **8** Department of Livestock Sciences and Techniques, University Sine Saloum El Hadji Ibrahima NIASS, Kaffrine, Senegal, **9** Department of Parasitology, Noguchi Memorial Institute for Medical Research, University of Ghana, Accra, Ghana, **10** Arbovirus/Viral Hemorrhagic Fevers Laboratory, Center for Virus Research, Kenya Medical Research Institute, Nairobi, Kenya, **11** Department of Solid Waste Management, Mosquito Control Division, Miami, Florida, United States of America, **12** Department of Ecology & Evolutionary Biology, Princeton University, Princeton, New Jersey, United States of America, **13** Princeton Neuroscience Institute, Princeton University, Princeton, New Jersey, United States of America

¤ Current address: Department of Ecology, Behavior, and Evolution, University of California San Diego, La Jolla, California, United States of America
* louis.lambrechts@pasteur.fr

## Abstract

African populations of the mosquito *Aedes aegypti* are usually considered less susceptible to infection by human-pathogenic flaviviruses than globally invasive populations found outside Africa. Although this contrast has been well documented for Zika virus (ZIKV), it is unclear to what extent it is true for dengue virus (DENV), the most prevalent flavivirus of humans. Addressing this question is complicated by substantial genetic diversity among DENV strains, most notably in the form of four genetic types (DENV1 to DENV4), that can lead to genetically specific interactions with mosquito populations. Here, we carried out a survey of DENV susceptibility using a panel of seven field-derived *Ae. aegypti* colonies from across the African range of the species and a colony from Guadeloupe, French West Indies as non-African reference. We found considerable variation in the ability of African *Ae. aegypti* populations to acquire and replicate a panel of six DENV strains spanning the four DENV types. Although African *Ae. aegypti* populations were generally less susceptible than the reference non-African population from Guadeloupe, in several instances some African populations were equally or more susceptible than the Guadeloupe population. Moreover, the relative level of susceptibility between African mosquito populations depended on the DENV strain, indicating genetically specific interactions. We conclude that unlike ZIKV

Agence Nationale de la Recherche (grant ANR-18-CE35-0003-01 to LL), the French Government's Investissement d'Avenir program Laboratoire d'Excellence Integrative Biology of Emerging Infectious Diseases (grant ANR-10-LABX-62-IBEID to LL), the US National Institutes of Health (grant NIDCD R00-DC012069 to CSM), and a Helen Hay Whitney Postdoctoral Fellowship (to NHR). CSM is a New York Stem Cell Foundation – Robertson Investigator. The funders had no role in study design, data collection and interpretation, or the decision to submit the work for publication.

**Competing interests:** The authors have declared that no competing interests exist.

susceptibility, there is no clear-cut dichotomy in DENV susceptibility between African and non-African *Ae. aegypti*. DENV susceptibility of African *Ae. aegypti* populations is highly heterogeneous and largely governed by the specific pairing of mosquito population and DENV strain.

## Author summary

African populations of the mosquito *Aedes aegypti* are usually thought to be less likely to get infected by flaviviruses compared to *Ae. aegypti* mosquitoes found outside Africa. While this has been well-demonstrated for Zika virus, it is not clear if the same is true for dengue virus, which is the most common flavivirus in humans. Studying this is complicated by the strain diversity of dengue virus, including four main genetic types, potentially causing different interactions. In this study, we compared several mosquito populations and found that, in general, African mosquitoes were less likely to get infected by dengue virus compared to mosquitoes from outside Africa. However, in some cases, African mosquitoes were just as or even more likely to get infected. The specific strain of dengue virus also influenced how likely African mosquitoes were to get infected, showing that the relationship between African mosquitoes and dengue virus is complex.

## Introduction

The mosquito *Aedes aegypti* is the main vector of several arthropod-borne viruses (arboviruses) of medical significance such as the flaviviruses dengue virus (DENV), Zika virus (ZIKV) and yellow fever virus (YFV). The species is native to Africa, but it is currently found throughout tropical and subtropical regions of the globe, and its distribution is expected to further expand in response to accelerating urbanization, connectivity, and climate change [1]. Two distinct subspecies of *Aedes aegypti* (that may even be considered distinct species [2]) were described by early taxonomists based on morphological and ecological differences [3] that were later associated with genetic variation [4]. *Aedes aegypti formosus* (*Aaf*) is a dark-colored, generalist subspecies found exclusively in sub-Saharan Africa that breeds both in forest and urban habitats and blood feeds on a variety of vertebrate hosts. *Aedes aegypti aegypti* (*Aaa*) is a light-colored, human-specialist subspecies found primarily outside Africa that preferentially bites humans and breeds in human-associated habitats. The dichotomy between *Aaf* and *Aaa* breaks down in some locations of Africa where genetically "admixed" populations are observed that display intermediate morphological and behavioral phenotypes [4–7].

The *Aaf* subspecies is considered a less efficient arbovirus vector than *Aaa* not only because of its lower affinity for human blood meals, but also because of a lower susceptibility to flavivirus infection [8]. Early comparative surveys reported a lower susceptibility to YFV [9,10] and DENV [11,12] of *Aaf* relative to *Aaa* populations. More recently, *Aaa* and *Aaf* were shown to differ significantly in ZIKV susceptibility [13]. Furthermore, the level of ZIKV susceptibility was found to correlate positively with the proportion of *Aaa* ancestry among African populations with varying levels of genetic admixture [13]. The genetic basis underlying variation in ZIKV susceptibility is not fully resolved but primarily lies in quantitative trait loci located on chromosome 2 [13]. Importantly, *Aaf* populations are less susceptible to ZIKV than *Aaa* populations irrespective of the virus strain [13].

Assessing arbovirus susceptibility in mosquitoes can be complicated by virus strain-specificity. Mosquito infection phenotypes are often determined by the specific pairing of the mosquito population and the virus strain, referred to as genotype-by-genotype (G x G) interactions [14]. G x G interactions are well documented for DENV susceptibility in *Ae. aegypti* [15–20], but these earlier studies mainly considered *Aaa* populations and did not directly compare *Aaa* and *Aaf*. It is unknown to what extent G x G interactions may challenge the universally lower DENV susceptibility of *Aaf* that was previously inferred from a limited number of DENV and mosquito strains. DENV exhibits substantial genetic diversity, most notably in the form of four genetic types (DENV1, DENV2, DENV3, and DENV4) that loosely cluster antigenically and are often referred to as serotypes [21]. In a recent study, we described an *Ae. aegypti* population from Bakoumba, Gabon displaying differential susceptibility to DENV3 and DENV1, resulting in significant G x G interactions when compared to a population from Cairns, Australia [22]. G x G interactions were also previously observed between several Senegalese *Ae. aegypti* populations and different flaviviruses [23]. Significant genetic variability between *Aaf* populations [5,7] and high genetic diversity of circulating DENV strains [24] makes it critical to account for both levels of variation when assessing DENV susceptibility. Here, we investigated continent-wide variation in DENV susceptibility across seven African *Ae. aegypti* populations using a panel of six African DENV strains spanning the four DENV types.

## Results

We used a panel of seven field-derived *Ae. aegypti* colonies (Table 1) from across the African range of the species and included a colony from Guadeloupe, French West Indies as a 100% *Aaa* reference. The panel of African DENV strains (Table 2) consisted of six wild-type viruses originally isolated from human serum from the four DENV types. In each experiment, the eight mosquito colonies were challenged simultaneously with one of the six DENV strains (3 increasing infectious doses each). The percentage of infected mosquitoes was determined by RT-PCR detection of viral RNA in mosquito bodies 12 days post infectious blood meal. In total, we tested 2,903 individual mosquitoes.

We first analyzed the proportion of infected mosquitoes as a function of virus dose (blood meal titer), DENV strain and mosquito population of origin (Fig 1). We analyzed infection prevalence by logistic regression, excluding the reference non-African population from Guadeloupe because we were primarily interested in the variation among African populations. Infection prevalence depended on a three-way interaction between virus dose, DENV strain and mosquito population, indicating that the dose-response curves differed significantly among virus-population pairs (Table 3). Dose-response curves account for the strong dose

**Table 1. Panel of *Ae. aegypti* colonies.** The average percentage of *Aaa* genetic ancestry (% *Aaa*) of each colony was determined based on whole-genome sequencing of their wild-caught progenitors [6,7].

| Locality of origin | Year of colonization | Lab generations | % *Aaa* |
| --- | --- | --- | --- |
| Saint François, Guadeloupe | 2015 | 20–23 | 100 |
| Praia, Cape Verde | 2020 | 4–7 | 23.0 |
| Ngoye, Senegal | 2018 | 12–14 | 37.4 |
| Ouagadougou, Burkina Faso | 2018 | 8–11 | 8.75 |
| Kumasi, Ghana | 2018 | 11–13 | 6.75 |
| Lopé, Gabon | 2014 | 24–27 | 7.30 |
| Entebbe, Uganda | 2015 | 19–22 | 0.00 |
| Rabai, Kenya | 2017 | 13–15 | 7.36 |

**Table 2. Panel of DENV strains.** All virus strains were originally isolated from human serum. The passage number refers to the number of amplifications in C6/36 cells prior to the experiments.

| Virus strain | DENV type | Locality of origin | Year of isolation | Passage number | Reference |
| --- | --- | --- | --- | --- | --- |
| DENV1_Gabon2010 | DENV1 | Franceville, Gabon | 2010 | 8 | [30] |
| DENV1_Somalia2012 | DENV1 | Somalia | 2012 | 4 | None |
| DENV2_Gabon2007 | DENV2 | Libreville, Gabon | 2007 | 3 | [30] |
| DENV2_Gabon2010 | DENV2 | Franceville, Gabon | 2010 | 4 | [30] |
| DENV3_Gabon2010 | DENV3 | Moanda, Gabon | 2010 | 7 | [30] |
| DENV4_Senegal1983 | DENV4 | Senegal | 1983 | 10 | [41] |

dependency of infection prevalence and provide an absolute measure of susceptibility, which can be summarized by the 50% oral infectious dose ($OID_{50}$), that is the blood meal titer expected to infect 50% of blood-fed mosquitoes [6,13]. We obtained the $OID_{50}$ estimates from the logistic fit of the dose-response curves. Comparison of $OID_{50}$ estimates confirmed that the level of DENV susceptibility depended on the specific pairing of mosquito population and DENV strain (Fig 2).

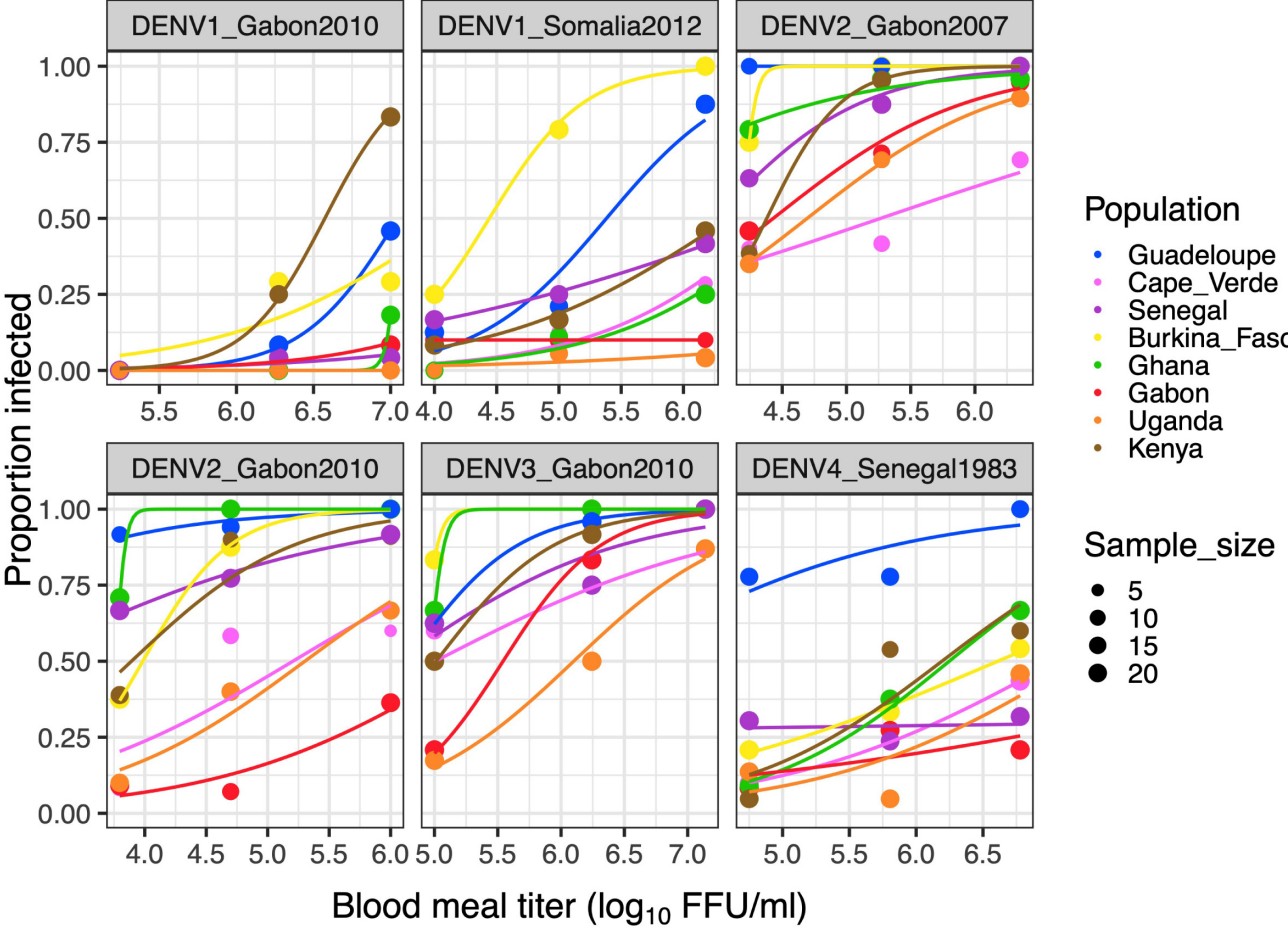

**Fig 1. Dose-response curves of infection prevalence for eight *Ae. aegypti* colonies challenged by six DENV strains.** The percentage of infected mosquitoes 12 days post oral challenge is shown as a function of the blood meal titer in $\log_{10}$ FFU/ml. Each panel represents a different DENV strain. A non-African colony from Guadeloupe, French West Indies is included as a 100% *Aaa* reference. Lines are logistic regressions of the data, color-coded by mosquito population. FFU: focus-forming units.

**Table 3. Test statistics of infection prevalence.** The table shows the result of a full-factorial logistic regression of infection status (excluding the reference Guadeloupe population).

| Variable | df | LR $\chi^2$ | p value |
|---|---|---|---|
| Virus strain | 5 | 687.2 | <0.0001 |
| Mosquito population | 6 | 105.1 | <0.0001 |
| Virus*Population | 30 | 125.2 | <0.0001 |
| Infectious dose | 1 | 0.001 | 0.9733 |
| Virus*Dose | 5 | 27.28 | <0.0001 |
| Population*Dose | 6 | 18.13 | 0.0059 |
| Virus*Population*Dose | 30 | 45.94 | 0.0315 |

df: degrees of freedom; LR: likelihood-ratio.

In general, the level of susceptibility of one mosquito population to a given DENV strain was not predictive of its susceptibility to another DENV strain (Fig 1). For example, *Ae. aegypti* from Ghana were among the most susceptible to the DENV2 and DENV3 strains, but they were among the most resistant to the DENV1 strains. Of note, the most susceptible mosquitoes were not always from the 100% *Aaa* population from Guadeloupe. For example, *Ae. aegypti* from Kenya were the most susceptible to the DENV1_Gabon2010 strain (Kenya: $OID_{50}$ = 6.57, 95% confidence interval [CI] = 6.34–6.79; Guadeloupe: $OID_{50}$ = 7.05, 95% CI = 6.82–7.61), and *Ae. aegypti* from Burkina Faso were the most susceptible to the DENV1_Somalia2012 strain (Burkina Faso: $OID_{50}$ = 4.44, 95% CI = 4.12–4.72; Guadeloupe: $OID_{50}$ = 5.35, 95% CI = 4.99–5.76). Only for the DENV4 strain were the mosquitoes from Guadeloupe significantly more susceptible than all the African mosquito populations. Overall, we did not detect a strong link between $OID_{50}$ estimates for different DENV strains or between $OID_{50}$ estimates and the proportion of *Aaa* genetic ancestry (S1 Fig), although our only unadmixed *Aaf* population (Uganda) showed consistently lower DENV susceptibility than admixed African populations. The DENV4 strain was the only one for which % *Aaa* was a significant predictor of $OID_{50}$ estimates (linear regression: $R^2$ = 0.59, $p$ = 0.044), but this relationship was largely driven by the 100% *Aaa* Guadeloupe population. We found no significant effect of the number of laboratory generations on the $OID_{50}$ estimates (linear regressions: $R^2$ = 0.01–0.51; $p$ = 0.175–0.828).

Next, we examined the level of systemic viral dissemination 12 days post oral challenge by quantifying infectious virus concentration in the head tissues of all infected mosquitoes (1,387 individuals in total) by end-point focus-forming assay. The prevalence of viral dissemination among infected mosquitoes was significantly influenced by the infectious dose and the virus-population pairing (Fig 3 and Table 4). Finally we analyzed non-zero dissemination titers, which are considered a proxy for DENV transmission potential because they are a strong predictor of the probability to detect infectious virus in mosquito saliva [25]. Dissemination titers were significantly influenced by the mosquito population and the interaction between infectious dose and DENV strain (Fig 4 and Table 5). Together, these results indicate that once infected, mosquitoes from different African populations also vary in their ability to disseminate DENV in a strain-specific manner.

## Discussion

Our survey of DENV susceptibility across seven African *Ae. aegypti* populations unveiled a more intricate relationship than previously presumed. Traditionally, African *Ae. aegypti* have been considered less likely to become infected by human-pathogenic flaviviruses compared to

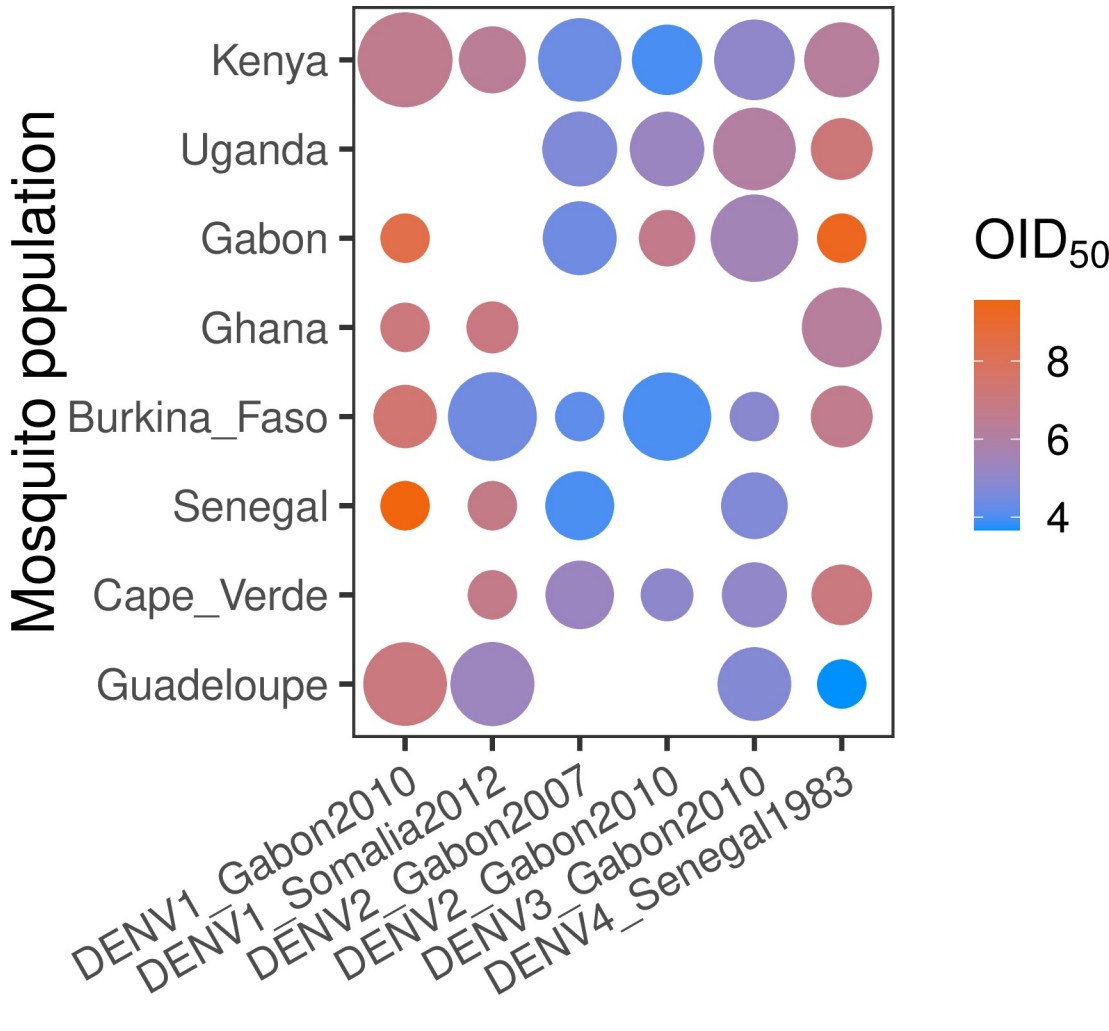

**Fig 2. OID$_{50}$ estimates for eight *Ae. aegypti* colonies challenged by six DENV strains.** OID$_{50}$ estimates are shown for each virus-population pair on a color scale in log$_{10}$ FFU/ml of blood. The size of the dot is inversely proportional to the size of the confidence interval of the OID$_{50}$ estimate. When the size of the confidence interval could not be estimated, it was arbitrarily set to 30 log$_{10}$ units. Lack of a dot means that the OID$_{50}$ could not be estimated with the data.

their counterparts outside of Africa [8]. This belief is supported by strong experimental evidence in the case of ZIKV [6,13], but the extension to other flaviviruses was not conclusively demonstrated. The present study challenges the notion of a clear-cut dichotomy in DENV susceptibility between African and non-African mosquitoes. Because we only included a single non-African *Ae. aegypti* population from Guadeloupe as an *Aaa* reference, our assessment is primarily relevant to compare African populations between them. Nevertheless, had there been a large phenotypic divergence in DENV susceptibility between *Aaa* and *Aaf*, the Guadeloupe population would have clearly stood out for all DENV strains.

In general, our findings were consistent with the initial assumption that African mosquito populations are generally less susceptible to DENV compared to non-African mosquitoes. Additionally, what emerged as an important insight was the substantial variation within African mosquito populations. While the majority exhibited lower susceptibility, some instances

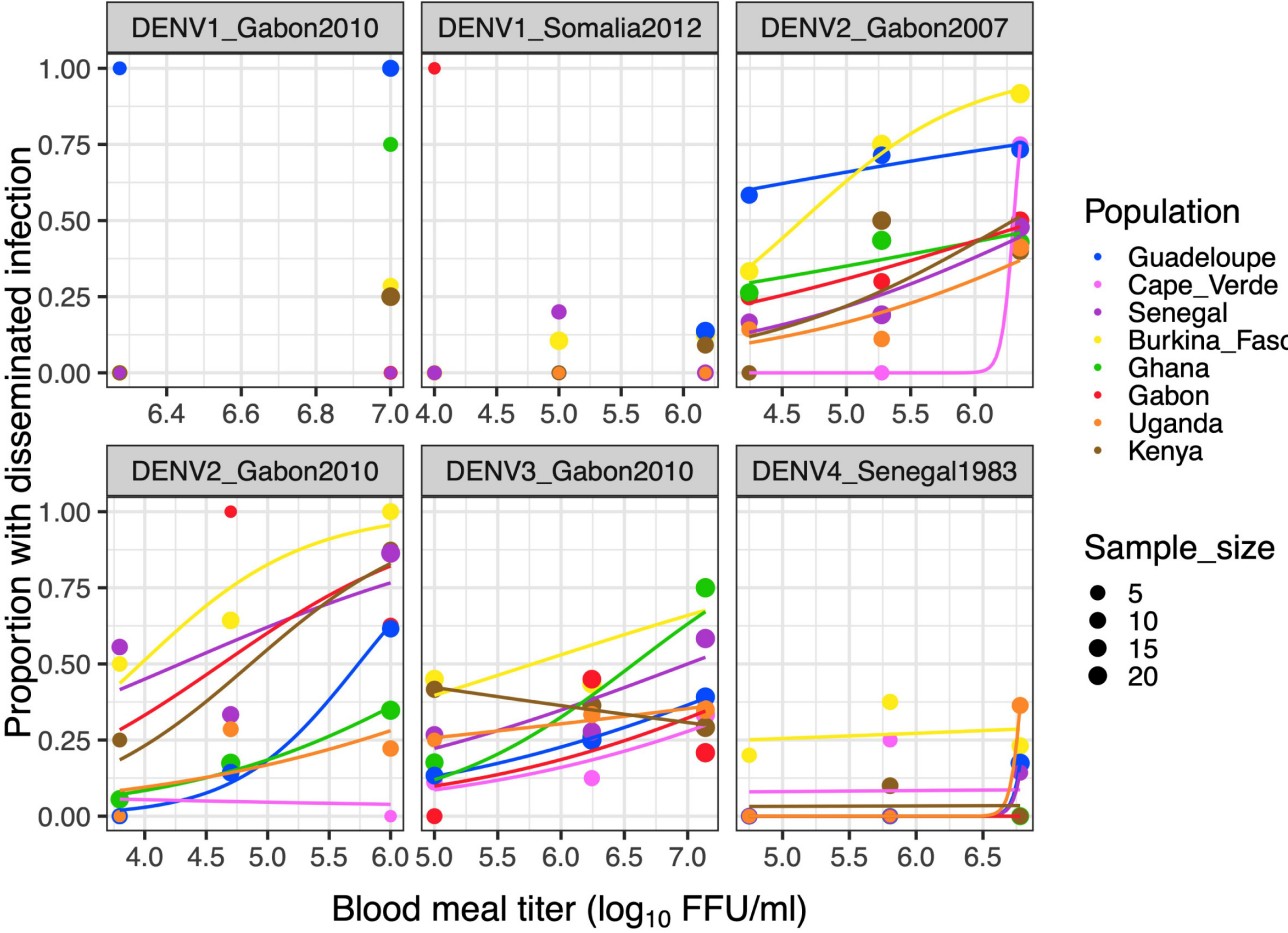

**Fig 3. Dose-response curves of viral dissemination prevalence for eight *Ae. aegypti* colonies challenged by six DENV strains.** The percentage of infected mosquitoes with viral dissemination to the head tissues 12 days post infection is shown as a function of the blood meal titer in $\log_{10}$ FFU/ml. Each panel represents a different DENV strain. A non-African colony from Guadeloupe, French West Indies is included as a 100% *Aaa* reference. Lines are logistic regressions of the data, color-coded by mosquito population. Logistic regression could not be performed for the DENV1 strains due to insufficient numbers of infected mosquitoes. FFU: focus-forming units.

revealed African populations that were equally or even more susceptible to DENV compared to the reference *Aaa* population from Guadeloupe. Although the relatively small number of admixed populations limited our statistical power to detect correlations, the genome-wide

**Table 4. Test statistics of dissemination prevalence.** The table shows the result of a full-factorial logistic regression of dissemination status among infected mosquitoes (excluding the reference Guadeloupe population). The DENV1 strains could not be included in this analysis due to insufficient numbers of infected mosquitoes that resulted in missing variable combinations. Non-significant terms were removed sequentially to obtain the minimal adequate model.

| Variable | df | LR $\chi^2$ | *p* value |
|---|---|---|---|
| Virus strain | 3 | 79.26 | <0.0001 |
| Mosquito population | 6 | 57.39 | <0.0001 |
| Virus*Population | 18 | 43.89 | 0.0006 |
| Infectious dose | 1 | 62.95 | <0.0001 |

df: degrees of freedom; LR: likelihood-ratio.

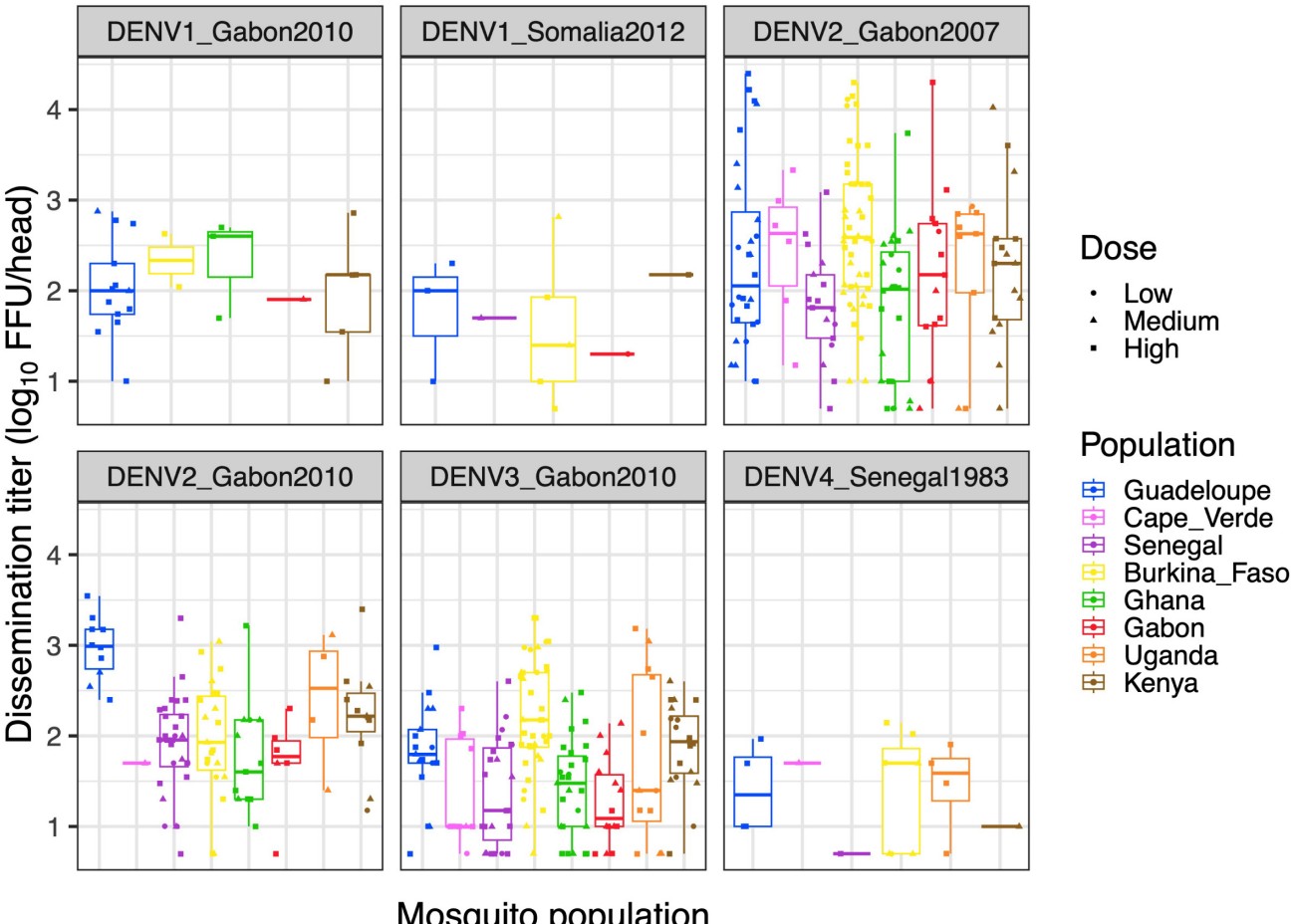

**Fig 4. Dissemination titers for eight *Ae. aegypti* colonies challenged by six DENV strains.** Boxplots show the $\log_{10}$-transformed distribution of non-zero infectious titers in the head tissues of mosquitoes with a disseminated infection (12 days post infectious blood meal) for each virus-population pair. The mosquito populations are color-coded, and symbols represent different doses (low, medium, and high blood meal titers). A non-African colony from Guadeloupe, French West Indies is included as a 100% *Aaa* reference. FFU: focus-forming units.

proportion of *Aaa* ancestry was not a reliable predictor of DENV susceptibility. However, it is possible that some of the phenotypic variation observed among admixed populations resulted from local ancestry effects, that is, variation in the proportion of *Aaa* ancestry at the specific loci that are relevant for DENV susceptibility.

**Table 5. Test statistics of non-zero dissemination titers.** The table shows the result of a full-factorial ANOVA of $\log_{10}$-transformed dissemination titers among mosquitoes with a disseminated infection (excluding the reference Guadeloupe population). The DENV1 and DENV4 strains could not be included in this analysis due to insufficient numbers of mosquitoes with a disseminated infection that resulted in missing variable combinations. Non-significant terms were removed sequentially to obtain the minimal adequate model.

| Variable | df | SS | F ratio | p value |
|---|---|---|---|---|
| Virus strain | 2 | 17.32 | 18.05 | <0.0001 |
| Mosquito population | 6 | 28.70 | 9.971 | <0.0001 |
| Infectious dose | 1 | 4.641 | 9.673 | 0.0020 |
| Virus*Dose | 2 | 3.235 | 3.372 | 0.0354 |
| Error | 350 | 225.3 | | |

df: degrees of freedom; SS: sum of squares.

Our results also highlighted the influential role of the specific virus strain in determining DENV susceptibility. The relationship between African mosquitoes and DENV was not a uniform phenomenon but rather a complex interplay influenced by both the mosquito population and the specific DENV strain. This confirms the pervasive nature of G x G interactions between DENV and *Ae. aegypti* for both *Aaa* and *Aaf* [15–20,22]. G x G interactions between hosts and pathogens are a prerequisite for local adaptation to occur [14,18]. For example, pathogen adaptation to vector populations has been observed between *Anopheles* mosquitoes and *Plasmodium* parasites [26]. Our experimental design did not allow testing for local adaptation patterns between DENV strains and *Ae. aegypti* populations because of the insufficient number of allopatric and sympatric combinations. An earlier study in Thailand did not provide support for DENV adaptation to local *Ae. aegypti* populations [16].

Our results may contribute to explain some unresolved features of dengue epidemiology in Africa. Dengue is present in several African countries, but its reported incidence is relatively lower compared to other regions like Southeast Asia and Latin America [27]. Large-scale dengue outbreaks in Africa have been less frequently documented compared to other continents, although sporadic cases and small outbreaks are detected regularly. For example, dengue outbreaks have been reported in Kenya [28,29], Gabon [30], Senegal [31,32], and Burkina Faso [33]. DENV prevalence in Africa might be more widespread that existing data suggest due to underreporting of dengue cases [34–36]. The disease may be misdiagnosed or underdiagnosed due to similarities in symptoms with other febrile illnesses, limited access to healthcare, and a lack of comprehensive surveillance systems. There are also indications that dengue is currently expanding in Africa [35]. Irrespective of the true prevalence of DENV, there is heterogeneity in its distribution between different regions of Africa [35,36]. The substantial variation and strain-specific patterns of DENV susceptibility among African *Ae. aegypti* observed in this study may contribute to explain this heterogeneity. Although more DENV4 strains are needed to conclusively address this possibility, it is tempting to speculate that the significantly lower DENV4 susceptibility of all African *Ae. aegypti* populations relative to the *Aaa* reference may be responsible, in part, for the rarity of DENV4 invasions in Africa [24].

Our findings also have implications for dengue prevention and control in Africa. The traditional assumption that African mosquitoes uniformly exhibit lower DENV susceptibility is challenged by the observed heterogeneity among mosquito populations. This diversity has significant consequences for the development of dengue prevention and control strategies in the region. A one-size-fits-all approach to dengue management may prove insufficient in the face of such variability. It could be more effective to tailor strategies based on the specific characteristics of the local mosquito populations and the prevalent DENV strains. For instance, regions with populations showing higher susceptibility to the circulating DENV strains may require more targeted and intensive vector control measures.

A limitation of our study is that the field-derived mosquito colonies had been maintained in the laboratory for up to 27 generations prior to the experiments (Table 1). Despite our efforts to maximize the number of reproducing adults at each generation during colony maintenance, laboratory adaptation and genetic drift are inevitable. Thus, it is possible that the colonies did not perfectly represent their populations of origin on the genetic level. Our results will need to be confirmed with mosquito colonies that are more recently derived from their wild progenitors.

In conclusion, this study challenges the conventional wisdom regarding DENV susceptibility in African *Ae. aegypti* and emphasizes the need for a nuanced and adaptive approach to dengue prevention and control in the region [37]. The complex interplay between mosquito populations and DENV strains adds a layer of intricacy that requires a thorough understanding for effective and targeted interventions. Understanding the factors influencing the

heterogeneous DENV susceptibility among African mosquito populations is the next critical step. It could involve exploring the genetic variations within *Ae. aegypti* populations in different regions, and the temporal dynamics of genetically specific interactions with DENV strains. Additionally, it will be important to assess how variation in DENV susceptibility combines with other parameters underlying vectorial capacity, such as human preference, to determine transmission risk [6].

## Methods

### Mosquitoes

Seven recently established *Ae. aegypti* colonies were chosen based on their geographical origins to best represent the African range of the species (Table 1). A colony from Guadeloupe, French West Indies was included as a non-African reference. Mosquitoes were reared under controlled insectary conditions (28˚C, 12h:12h light:dark cycle and 70% relative humidity). Prior to performing the experiments, their eggs were hatched synchronously in a vacuum chamber for 1 hour. Their larvae were reared in plastic trays containing 1.5 liter of dechlorinated tap water and supplemented with a standard diet of Tetramin (Tetra) fish food at a density of 200 larvae per tray. After emergence, adults were kept in $30 \times 30 \times 30$ cm BugDorm-1 insect cages (BugDorm) with permanent access to 10% sucrose solution. For each experiment, all the mosquito colonies were reared simultaneously in the same insectary.

### Viruses

Six wild-type DENV strains originally isolated from human serum in Africa were chosen to represent the four DENV genetic types (Table 2). Viruses were amplified in the C6/36 *Aedes albopictus* cell line (ATCC CRL-1660) to generate viral stocks as previously described [38]. The C6/36 wells were maintained at 28˚C under atmospheric $CO_2$ in tissue-culture flasks with non-vented caps, in Leibovitz's L-15 medium complemented with 10% fetal bovine serum (FBS), 2% tryptose phosphate broth (Gibco ThermoFisher Scientific), 1× non-essential amino acids (Gibco ThermoFisher Scientific), 10 U/ml of penicillin (Gibco Thermo Fisher Scientific) and 10 μg/ml of streptomycin (Gibco ThermoFisher Scientific). DENV infectious titers were measured in C6/36 cells using a standard focus-forming assay (FFA) as previously described [38]. A commercial mouse anti-DENV complex monoclonal antibody (MAB8705; Merck Millipore) diluted 1:200 in phosphate-buffered saline (PBS; Gibco Thermo Fisher Scientific) supplemented with 1% bovine serum albumin (BSA; Interchim) was used as the primary antibody. The secondary antibody was an Alexa Fluor 488-conjugated goat anti-mouse antibody (A-11029; Life Technologies) diluted 1:500 in PBS supplemented with 1% BSA.

### Experimental infections

Mosquitoes were orally challenged with DENV by membrane feeding as previously described [39]. Briefly, five- to seven-day-old females deprived of sucrose solution for 24 hours were offered an artificial infectious blood meal for 20 min using a Hemotek membrane-feeding apparatus (Hemotek Ltd.) with porcine intestine as the membrane. Blood meals consisted of a 2:1 mix of washed commercial rabbit erythrocytes (BCL) and virus suspension. To establish the dose responses, the mosquitoes were exposed to different virus concentrations by diluting the virus stocks in cell culture medium prior to preparing the artificial infectious blood meal. Adenosine triphosphate (Merck) was added to the blood meal as a phagostimulant at a final concentration of 10 mM. Fully engorged females were sorted on wet ice, transferred into 1-pint cardboard containers and maintained under controlled conditions (28˚, 12h:12h light:

dark cycle and 70% relative humidity) in a climatic chamber with permanent access to 10% sucrose solution. After 12 days of incubation, mosquitoes were cold anesthetized, and their head and body were separated from each other and stored at –80˚C. Infection prevalence was determined by RT-PCR of bodies, whereas viral dissemination titers were determined by FFA of heads from mosquitoes with a virus-positive body. RT-PCR is a reliable and sensitive assay to determine infection prevalence. FFA was used to quantify infectious virus in head tissues because dissemination titer is a proxy for DENV transmission potential [25].

## Sample testing

Head-less mosquito bodies were homogenized individually in 300 μl of squash buffer (Tris 10 mM, NaCl 50 mM, EDTA 1.27 mM with a final pH adjusted to 9.2) supplemented with proteinase K (Eurobio Scientific) at a final concentration of 0.35 mg/ml. The body homogenates were clarified by centrifugation and 100 μl of each supernatant were incubated for 5 min at 56˚C followed by 10 min at 98˚C to extract viral RNA. Detection of viral RNA was performed using a two-step RT-PCR reaction targeting a conserved region of the DENV *NS5* gene. Total RNA was reverse transcribed into cDNA using random hexameric primers and the M-MLV reverse transcriptase (ThermoFisher Scientific) by the following program: 10 min at 25˚C, 50 min at 37˚C and 15 min at 70˚C. The cDNA was subsequently amplified using DreamTaq DNA polymerase (ThermoFisher Scientific). For this step, 20-μl reaction volumes contained 1× of reaction mix and 10 μM of primers. Specific primer pairs were used to detect DENV1_Gabon2010 (Forward: 5'-CCGACTTGTCCACTTCCTCT-3'; Reverse: 5'-TTGG GAGCACGCTTTCTAGA-3'), DENV1_Somalia2012 (Forward: 5'-CGAAGATCACTGGTT CAGCA-3'; Reverse: 5'-ACATCCATCACGGTTCCATT-3'), both DENV2 strains (Forward: 5'-CGCTTCTTAGAGTTTGAAGCCC-3'; Reverse: 5'-GGTCTTTGCACACGCACC-3'), DENV3_Gabon2010 (Forward: 5'-AGAAGGAGAAGGACTGCACA-3'; Reverse: 5'- ACCT GTCCACTGCCTCTTTG-3') and DENV4_Senegal1983 (Forward: 5'- CTGGAATTTGAAG CCCTGGG-3'; Reverse: 5'-GGGTCTGAGGACTTTCACCA-3'). The thermocycling program was 2 min at 95˚C, 35 cycles of 30 sec at 95˚C, 30 sec at 60˚C, and 30 sec at 72˚C with a final extension step of 7 min at 72˚C. Amplicons were visualized by electrophoresis on a 2% agarose gel. Individual heads were homogenized in 200 μl of Leibovitz's L-15 medium with 2% TPB, 1× NAA, 10 U/ml of penicillin, and 10 μg/ml of streptomycin. Infectious titers were determined by end-point FFA in C6/36 cells as previously described [38].

## Statistics

Statistical analyses were performed in JMP v10.0.2 (www.jmpdiscovery.com), and graphical representations were generated with the R package *ggplot2* [40]. Prevalence (infection and dissemination) was analyzed by nominal logistic regression and likelihood-ratio $\chi^2$ tests. Non-zero dissemination titers were $\log_{10}$-transformed and compared by analysis of variance (ANOVA). The full-factorial model included infectious dose ($\log_{10}$-transformed blood meal titer), mosquito population and virus strain as covariates. Non-significant terms were removed sequentially to obtain the minimal adequate model. The $OID_{50}$ estimates and their respective 95% confidence intervals were derived from the logistic fits of infection prevalence.

## Supporting information

**S1 Table. Raw data of the study.**
(XLSX)

**S1 Fig. Correlations between DENV susceptibility levels for different virus strains and the percentage of *Aaa* ancestry.** The Pearson linear correlations between $OID_{50}$ estimates are shown for each pair of DENV strains and with % *Aaa* (rightmost column). The black lines represent the linear correlations, and the grey shading indicates their confidence interval. The mosquito populations are color-coded; their average % *Aaa* was determined based on whole-genome sequencing of their wild-caught progenitors.
(EPS)

## Acknowledgments

We thank Catherine Lallemand for assistance with mosquito rearing, Alexander Bergman for technical help with graphical displays, and the other members of the Lambrechts lab for their insights. We are grateful to Scott O'Neill for his suggestions. We thank Eric Leroy, Jean-Bernard Lekana-Douki, Isabelle Moltini-Conclois, Marie-Pascale Frenkiel, Albin Fontaine and Isabelle Leparc-Goffart for facilitating the transfer of virus strains.

## Author Contributions

**Conceptualization:** Louis Lambrechts.

**Data curation:** Louis Lambrechts.

**Formal analysis:** Louis Lambrechts.

**Funding acquisition:** Louis Lambrechts.

**Investigation:** Stéphanie Dabo, Annabelle Henrion-Lacritick, Alicia Lecuyer.

**Methodology:** Stéphanie Dabo.

**Project administration:** Louis Lambrechts.

**Resources:** Davy Jiolle, Christophe Paupy, Diego Ayala, Silvânia da Veiga Leal, Athanase Badolo, Anubis Vega-Rúa, Massamba Sylla, Jewelna Akorli, Sampson Otoo, Joel Lutomiah, Rosemary Sang, John-Paul Mutebi, Noah H. Rose, Carolyn S. McBride.

**Supervision:** Maria-Carla Saleh, Louis Lambrechts.

**Visualization:** Louis Lambrechts.

**Writing – original draft:** Louis Lambrechts.

**Writing – review & editing:** Stéphanie Dabo, Noah H. Rose, Carolyn S. McBride, Louis Lambrechts.

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
