## [Decision Letter · Decision Letter 0]

29 Jan 2024

Dear Dr. Lambrechts,

Thank you very much for submitting your manuscript "Extensive variation and strain-specificity in dengue virus susceptibility among African Aedes aegypti populations" for consideration at PLOS Neglected Tropical Diseases. As with all papers reviewed by the journal, your manuscript was reviewed by members of the editorial board and by several independent reviewers. In light of the reviews (below this email), we would like to invite the resubmission of a significantly-revised version that takes into account the reviewers' comments. 

We cannot make any decision about publication until we have seen the revised manuscript and your response to the reviewers' comments. Your revised manuscript is also likely to be sent to reviewers for further evaluation.

Sincerely,

Doug E Brackney

Academic Editor

Abdallah Samy

Section Editor

Editor comments to authors: 

The reviewers tended to like the study and had some minor improvements. Specifically, I tend to agree with Reviewer 2 that titer data should be calculated by removing the zero values. The proportion data tells us how many became infected whereas the titer data should tell us how well the virus is able to replicate in those mosquitoes that became infected. Also, Reviewer 1 suggested adding some caveats to the discussion about the potential that inbreeding of the colonies and passaging of the virus could have on the outcomes. Reviewer 3 pointed out some useful citations concerning DENV activity in Africa. Some text should be added to the manuscript regarding DENV activity in Africa. Reviewer 3 also questioned the conclusions about transmission potential of your disseminated mosquitoes. Additional studies are not required to address this but justifying your conclusions with a reference(s) would be recommended.

Reviewer's Responses to Questions

**Key Review Criteria Required for Acceptance?**

**Methods**

-Are the objectives of the study clearly articulated with a clear testable hypothesis stated?

-Is the study design appropriate to address the stated objectives?

-Is the population clearly described and appropriate for the hypothesis being tested?

-Is the sample size sufficient to ensure adequate power to address the hypothesis being tested?

-Were correct statistical analysis used to support conclusions?

-Are there concerns about ethical or regulatory requirements being met?

Reviewer #1: Below I pointed the issues related to mosquito lines and inbreeding and DENV isolates (time) which could have impacted the results.

Reviewer #2: Yes

Reviewer #3: The objectives of the study are clearly articulated, however, a few areas need to be clarified or discussed:

What were the individual DENV blood meal titers used to infect each mosquito population in order to come up with the OID50?

Why were 2 different methods or approaches used to determine the infection and dissemination rates? That is PCR to determine infection rates and FFA to determine the dissemination rates.

**Results**

-Does the analysis presented match the analysis plan?

-Are the results clearly and completely presented?

-Are the figures (Tables, Images) of sufficient quality for clarity?

Reviewer #1: Yes

Reviewer #2: Yes

Reviewer #3: To create clarity and allow the reader to interpret the results better, other than the graphs, it will be important to provide a table that provides raw figures of the mosquitoes exposed to each viral titer, the percentage infected and the percentage disseminating the virus. 

It may be an overreach to state that dissemination can be considered as transmission potential as many studies have shown high levels of dissemination into the body but lack of transmission of the virus in expectorated saliva. It may be only safe to consider pathogens present in the salivary gland as that likely to be transmitted to the saliva. 

It may be important to explain why the Guadaloupe reference strain was excluded during the dissemination analysis 

Line 174 and 175 should be altered to demonstrate what you're study investigated as you investigated dissemination but not transmission.

**Conclusions**

-Are the conclusions supported by the data presented?

-Are the limitations of analysis clearly described?

-Do the authors discuss how these data can be helpful to advance our understanding of the topic under study?

-Is public health relevance addressed?

Reviewer #1: Partially

Reviewer #2: Yes

Reviewer #3: It will be important to include in your discussion the effect of using a low lab generation of Cape Verde on the overall results. 

Lines 179-180: This might need to be re-worded as there have been contradictory findings on the vector competence of Aaf for DENV and YFV

There have also been reports of several Dengue outbreaks in African countries driven by local populations of Aedes aegypti, hence there is a possibility that African populations of Aedes aegypti are susceptible to Dengue virus

Dickson, L. B., Sanchez-Vargas, I., Sylla, M., Fleming, K. & Black, W. C., 4th. Vector competence in West African Aedes aegypti Is Flavivirus species and genotype dependent. PLoS Negl. Trop. Dis. 8, e3153 (2014).

Tabachnick, W. J. et al. Oral infection of Aedes aegypti with yellow fever virus: geographic variation and genetic considerations. Am. J. Trop. Med. Hyg. 34, 1219–1224 (1985).

Bosio, C. F., Beaty, B. J. & Black, W. C., 4th. Quantitative genetics of vector competence for dengue-2 virus in Aedes aegypti. Am. J. Trop. Med. Hyg. 59, 965–970 (1998).

Lines 216-217 may need to be re-worded

There have been reports of several Dengue outbreaks in African countries that have been documented in the past: This is just a single country:

Johnson, B. K. et al. Epidemic dengue fever caused by dengue type 2 virus in Kenya: preliminary results of human virological and serological studies. East Afr. Med. J. 59, 781–784 (1982).

Konongoi, L. et al. Detection of dengue virus serotypes 1, 2 and 3 in selected regions of Kenya: 2011-2014. Virol. J. 13, 182 (2016).

Gathii, K., Nyataya, J. N., Mutai, B. K., Awinda, G. & Waitumbi, J. N. Complete Coding Sequences of Dengue Virus Type 2 Strains from Febrile Patients Seen in Malindi District Hospital, Kenya, during the 2017 Dengue Fever Outbreak. Genome Announc. 6, (2018).

Muthanje, E. M. et al. March 2019 dengue fever outbreak at the Kenyan south coast involving dengue virus serotype 3, genotypes III and V. PLOS Global Public Health vol. 2 e0000122 Preprint at https://doi.org/10.1371/journal.pgph.0000122

**Editorial and Data Presentation Modifications?**

Reviewer #1: (No Response)

Reviewer #2: (No Response)

Reviewer #3: N/A

**Summary and General Comments**

Reviewer #1: The present study performed by Dabo and colleagues studied the vector competence of African Aedes aegypti populations towards DENV isolated from Africa. The manuscript is well written, and the methods, although simplified, were appropriate for the topic in question. The work is quite simple and had no major link around the vector competence of African mosquito populations and DENV, maybe because of the points discussed below.

Major points

Mosquito populations: 5-25 generations in the lab means a lot of inbreeding and might not represent the field component of the mosquito line and origin.

Virus isolates: Isolates have been collected quite long ago and multiple passaging could impact on the infectivity. It was not present on the methods section how these isolates have been kept (whether in constant passaging, which could impact mosquito infectivity; or as frozen stocks). Also, it was a pity that there were no respective virus isolates to match each mosquito population. Most of the isolates came from Gabon.

Including a discussion around these caveats would be beneficial to the study.

Reviewer #2: The manuscript by Dabo et al. aims to measure whether African populations of Ae. aegypti are more refractory to DENV compared to non-African populations. The authors compare the vector competence of panel of Ae. aegypti lines from within Africa to a reference colony collected outside Africa. Using a panel of different DENV serotypes and genotypes the authors clearly demonstrates that unlike with ZIKV, the vector competence of the African lines is not lower than a line from Guadeloupe in all cases. Instead, vector competence depends on the specific pairing between mosquito population and DENV strain. The manuscript is very well written and the data clearly supports the conclusions. 

Minor Comments:

Line 45: Seven colonies is hardly a continent-wide survey. Language should be softened.

Figure 3: This figure is hard to read. Zeroes should not be included when comparing the titers between individuals. It is more informative to compare titers of those individuals who are infected and then separately plot the percentage of individuals with a disseminated infection.

Reviewer #3: The authors may need to provide a bit more clarity with regard to their data presentation as well as update and highlight other findings and publications about Dengue circulation and outbreaks in several countries in Africa which may explain some of their findings. This will bring clarity and improve their discussion sections. Comments on specific sections have been provided above.

PLOS authors have the option to publish the peer review history of their article (what does this mean?). If published, this will include your full peer review and any attached files.

Reviewer #1: No

Reviewer #2: No

Reviewer #3: No
---

## [Editor Report · Decision Letter 1]

15 Mar 2024

Dear Dr. Lambrechts,

We are pleased to inform you that your manuscript 'Extensive variation and strain-specificity in dengue virus susceptibility among African Aedes aegypti populations' has been provisionally accepted for publication in PLOS Neglected Tropical Diseases.

Best regards,

Doug E Brackney, PhD

Academic Editor

Abdallah Samy

Section Editor

---

## [Editor Report · Acceptance letter]

19 Mar 2024

Dear Dr. Lambrechts,

We are delighted to inform you that your manuscript, "Extensive variation and strain-specificity in dengue virus susceptibility among African *Aedes aegypti* populations," has been formally accepted for publication in PLOS Neglected Tropical Diseases.

Best regards,

Shaden Kamhawi

co-Editor-in-Chief

Paul Brindley

co-Editor-in-Chief
